# ProGAE: A Geometric Autoencoder-based Generative Model for Disentangling Protein Conformational Space

## Abstract

Understanding the protein conformational landscape is critical, as protein function, as well as modulations thereof due to ligand binding or changes in environment, are intimately connected with structural variations. This work focuses on learning a generative neural network on a simulated ensemble of protein structures obtained using molecular simulation to characterize the distinct structural fluctuations of a protein bound to various drug molecules. Specifically, we use a geometric autoencoder framework to learn separate latent space encodings of the intrinsic and extrinsic geometries of the system. For this purpose, the proposed Protein Geometric AutoEncoder (ProGAE) model is trained on the length of the alpha-carbon pseudobonds and the orientation of the backbone bonds of the protein. Using ProGAE latent embeddings, we reconstruct and generate the conformational ensemble of a protein at or near the experimental resolution. Empowered by the disentangled latent space learning, the intrinsic latent embedding help in geometric error correction, whereas the extrinsic latent embedding is successfully used for classification or property prediction of different drugs bound to a specific protein. Additionally, ProGAE is able to be transferred to the structures of a different state of the same protein or to a completely different protein of different size, where only the dense layer decoding from the latent representation needs to be retrained. Results show that our geometric learning-based method enjoys both accuracy and efficiency for generating complex structural variations, charting the path toward scalable and improved approaches for analyzing and enhancing molecular simulations.

## 1 Introduction

The complex and time-consuming calculations in molecular simulations have been significantly impacted by the application of machine learning techniques in recent years. In particular, deep learning has been applied to analysis and simulation of molecular trajectories to address diverse problems, such as estimating free energy surfaces, defining optimal reaction coordinates, constructing Markov State Models, and enhancing molecular sampling. For a comprehensive review of deep learning methods for analyzing and enhancing molecular simulations, see (Noé et al., 2020a) and (Noé et al., 2020b).

Specifically, there has been interest in modeling the underlying conformational space of proteins by using deep generative models, e.g. (Ramaswamy et al., 2020) and (Bhowmik et al., 2018; Guo et al., 2020; Varolgüneş et al., 2020). This line of work has mainly attempted to respect the domain geometry by using convolutional AEs on features extracted from 3D structures. In parallel, learning directly from 3D structure has recently developed into an exciting and promising application area for deep learning. In this work, we learn the protein conformational space from a set of protein simulations using geometric deep learning. We also investigate how the geometry of a protein itself can assist learning and improve latent conformational space interpretability. Namely, we consider the influence of intrinsic and extrinsic geometry, where intrinsic geometry is independent of 3D embedding and extrinsic is not. Intrinsic geometric protein properties can be thought to be robust to conformation. To this end, we propose a Protein Geometric Autoencoder model, named ProGAE, to separately encode intrinsic and extrinsic protein geometries.

The main contributions of this work are summarized:

- Inspired by recent unsupervised geometric disentanglement learning works (Tatro et al., 2020; Wu et al., 2019; Yang et al., 2020), we propose a novel geometric autoencoder named ProGAE that directly learns from 3D protein structures via separately encoding intrinsic and extrinsic geometries into disjoint latent spaces used to generate protein structures.

- We further propose a novel formulation, in which network intrinsic input is taken as the $C_\alpha$-$C_\alpha$ pseudo-bond distances, and the extrinsic input is the backbone bond orientations.

- Analysis shows that the learned extrinsic geometric latent space can be used for drug classification and drug property prediction, where the drug is bound to the given protein.

- We find that the intrinsic geometric latent space, even with small variation in the intrinsic input signal, is important for reducing geometric errors in reconstructed proteins.

- We also demonstrate that the learned ProGAE can be transferred to a trajectory of the protein in a different state or a trajectory of a different protein all-together.

## 1.1 RELATED WORK

Recently, a body of work has used deep learning to learn from protein structures (Graves et al., 2020; Jing et al., 2020; Klicpera et al., 2020). For example, Gainza et al. (2019) uses geometric deep learning to predict docking sites for protein interactions. Ingraham et al. (2019a) solves the *inverse folding problem* using a graph transformer on the protein backbone. Degiacomi (2019) uses an AE to generate candidate proteins for docking. Hermosilla et al. (2020) leverages the notion of intrinsic and extrinsic geometry to define an architecture for a fold classification task.

Additionally, there has been focus on directly learning the temporal aspects of molecular dynamics from simulation trajectories, which is not directly related to the current work. Please see Appendix A.1 for a detailed discussion.

There is an existing body of recent works that use AE-based approaches for either analyzing and/or generating structures from from the latent space (Bhowmik et al., 2018; Guo et al., 2020; Ramaswamy et al., 2020; Varolgüneş et al., 2020), which are most closely related to this work. (Bhowmik et al., 2018) and (Guo et al., 2020) aim at learning from and generating protein contact maps, while ProGAE directly deals with 3D structures. Therefore a direct comparison of ProGAE with these methods is not possible. Ramaswamy et al. (2019) uses a 1D CNN autoencoder trained on backbone coordinates and uses a loss objective comprised of geometric MSE error and physics-based (bond length, bond angle, etc.) error. Due to the unavailability of code or pre-trained model, we were unable to perform a direct comparison. Varolgüneş et al. (2020) uses a VAE with a Gaussian Mixture Prior for performing clustering of high-dimensional input configurations in the learned latent space. While the method works well on toy models and a standard Alanine Dipeptide benchmark, its performance drops as the size of the protein system grows to 15 amino acids, which is approximately an order smaller than the protein systems studied here. Also, their approach is likely not going to scale well to larger systems due to the use of fully-connected layers in the encoder.

These mentioned works have not considered explicit disentangling of intrinsic and extrinsic geometries. To our knowledge, this work is the first to propose an autoencoder for the unsupervised modeling of the geometric disentanglement of protein conformational space captured in molecular simulations. This representation provides better interpretability of the latent space, in terms of the physico-chemical and geometric attributes, results in more geometrically accurate protein conformations, as well as scales and transfers well to larger protein systems.

## 2 PROGAE FOR PROTEIN CONFORMATIONAL SPACE

First, we introduce the input signal for our novel geometric autoencoder, ProGAE. We then discuss how ProGAE utilizes this signal to generate the conformational space of a protein.

**Geometric Features of Protein as Network Input** ProGAE functions by separately encoding intrinsic and extrinsic geometry with the goal of achieving better latent space interpretability. We

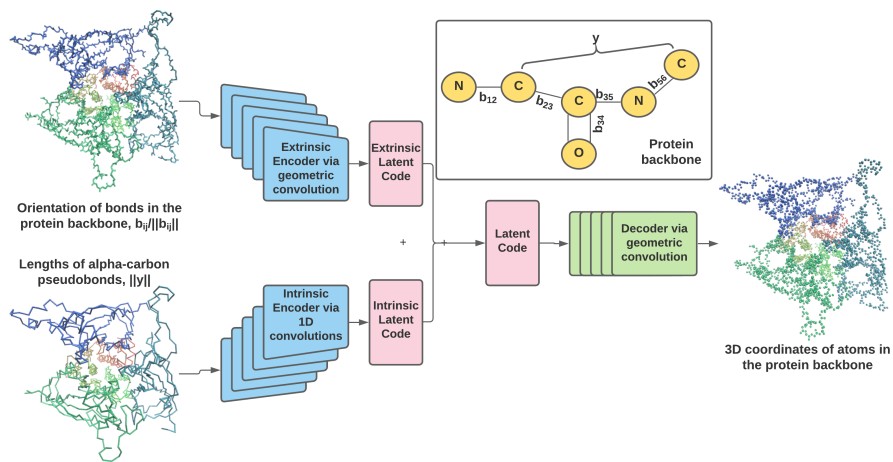

Figure 1: Architecture of our network, ProGAE, that generates protein conformations via separate encoding of data related to coarse intrinsic and extrinsic geometries. These geometries are captured via the orientation of the backbone bonds (extrinsic) and length of $C_\alpha - C_\alpha$ pseudobonds (intrinsic). These latent representations are jointly used to generate the 3D coordinates of the backbone atoms.

clarify these geometric notions. Mathematically, we can consider a manifold (i.e. surface) independent of its embedding in Euclidean space. Properties that do not depend on this embedding are known as intrinsic geometric properties, while properties that do are referred to as extrinsic. As an example, given two atoms of a protein, the intrinsic distance between them is the minimum sum of bond lengths in the bond path connecting them, whereas the extrinsic distance is their Euclidean distance in $\mathbb{R}^3$. For an in-depth review of geometry, we refer the reader to (Do Carmo, 2016).

As we will train ProGAE to learn the conformational space of a given protein, the protein primary structure is implicit. Then in treating it as a geometric object, we view the protein at the level of its backbone, which specifies its shape. Given primary structure, reconstructing the protein backbone is sufficient for reconstructing the entire protein. Of importance in the backbone are the $C_\alpha$ atoms, which are the centers of amino acids in the protein. Then a coarse-level description of the backbone is the $C_\alpha$ atoms connected linearly in terms of the protein sequence. This is known as the trace of the protein. We will use the backbone and trace as domains on which to define our signals.

Both the protein backbone and its trace can be viewed as polygonal chain in Euclidean space. They are depicted in Figure 1 with their geometric features as network input. We can see that a polygonal chain can be determined up to translation given both the length and orientation of its line segments. Then it follows that the protein backbone can be determined given the length and orientation of its bonds. Here the length of these bonds is intrinsic while the orientation is extrinsic. Thus, to decouple the intrinsic and extrinsic geometry, we can consider encoding these signals.

The length of covalent bonds undergo very little change during a simulation performed using an empirical force-field, like the simulations considered in this work. A standard deviation of less than $0.059$Å from target bond lengths is common in PDB structures (Jaskolski et al., 2007). To this end, we instead consider intrinsic geometry at a coarse level, so that the resulting signal has more variability. Specifically, we use length of the $C_\alpha$-$C_\alpha$ pseudobonds in the trace as a representative of the intrinsic protein geometry, where as backbone bond orientations capture extrinsic geometry.

We model the backbone by the graph, $\mathcal{G}_b = (\mathbb{V}_b, \boldsymbol{E}_b)$, and the backbone trace by the graph, $\mathcal{G}_t = (\mathbb{V}_t, \boldsymbol{E}_t)$. Then our intrinsic and extrinsic signals, $Int : \boldsymbol{E}_t \to \mathbb{R}$ and $Ext : \boldsymbol{E}_b \to \mathbb{R}^3$ are defined:

$$Int(E_{ij}) = \|E_{ij}\|_2, \quad E_{ij} \in \boldsymbol{E}_t, \qquad Ext(E_{ij}) = sgn(j - i)\frac{E_{ij}}{\|E_{ij}\|}, \quad E_{ij} \in \boldsymbol{E}_b. \qquad (1)$$

**Network Architecture** With the network inputs defined, we discuss the architecture of ProGAE. The core idea is to create an *intrinsic* latent space, $L_I \in \mathbb{R}^{n_i}$, and an *extrinsic* latent space, $L_E \in \mathbb{R}^{n_e}$, via separately encoding the intrinsic and extrinsic signals. Consequently, our network contains

two encoders, $Enc_i$ and $Enc_e$ where:

$$Enc_i \circ Int(\boldsymbol{E}_t) \in L_I, \qquad Enc_e \circ Ext(\boldsymbol{E}_b) \in L_E. \qquad (2)$$

We then jointly decode these latent vectors to recover the coordinates of the atoms in the protein backbone. Thus, we formally define the decoder:

$$Dec : L_I \times L_E \to \mathbb{R}^{|\mathbb{V}_b| \times 3}. \qquad (3)$$

This high level structure of ProGAE is depicted in Figure 1. We provide additional details on the encoders and decoders. As these edge-based signals are defined on a geometric domain, it is sensible to learn feature representations using geometric convolution that respects the geometry of the data. The intrinsic encoder is simple, as the signal is defined on the backbone trace, which corresponds to a set of discrete curves. Here each curve corresponds to a protein fragment. Then we define $Enc_i$ to be a series of 1D convolutions operating on each protein fragment. Each convolution is taken to have a kernel size of 3 and a stride of 2, being followed with batch normalization layers and ReLU.

In contrast, the extrinsic encoder operates on the backbone, which we associate with a graph. So the layers of graph attention networks (GATs) introduced in (Veličković et al., 2017) are a natural tool to use, albeit with some modification. Since the input signal is defined only on the edges of the graph, $\boldsymbol{E}_b$, we define a signal on the graph vertices, $\mathbb{V}_b$, as the average value of its incident edges,

$$f_0(v_i) := \frac{1}{|\{j; E_{i,\cdot} \in \boldsymbol{E}_b\}|} \sum_{j; E_{i,\cdot} \in \boldsymbol{E}_b} Ext(E_{ij}), \quad v_i \in \mathbb{V}_b. \qquad (4)$$

Then the first layer of the extrinsic encoder uses the edge-convolution operator of (Gong & Cheng, 2019) to map this graph signal to a signal defined exclusively on the graph vertices, $\mathbb{V}_b$. The rest of the encoder contains successive graph attention layers with sparsity defined by a given neighborhood radius. At each layer, the signal is downsampled by a factor of two based on farthest point sampling. Given $L$ layers, this defines a sequence of graphs, $\{\mathcal{G}_{b,i}\}_{i=0}^L$, with increasing decimation. As with $Enc_i$, each layer is followed with batch normalization and ReLU. Summarily, for $l = 1, 2, ..., L$,

$$f_l = \sigma \circ BN \circ GAT(d_{l-1}) \text{ where } d_{l-1} = DS(f_{l-1}; 2), f_1(v_i) := GAT(f_0(\mathbb{V}_b), Ext(\boldsymbol{E}_b)). \qquad (5)$$

Global average pooling is applied to the encoder outputs to introduce invariance to size of $\mathbb{V}_t$ and $\mathbb{V}_b$. Dense layers then map each result to their respective latent spaces, $L_I$ and $L_E$. The Tanh function is applied to bound the latent space. This produces the intrinsic and extrinsic latent codes, $\boldsymbol{z}_i$ and $\boldsymbol{z}_e$.

The latent code $\boldsymbol{z}$ is taken as the concatenation of the two latent codes, $[\boldsymbol{z}_i, \boldsymbol{z}_e]$. A dense layer maps $\boldsymbol{z}$ to the a signal defined on the most decimated backbone graph, $\mathcal{G}_{b,L}$. The structure of the decoder, $Dec$, is then analogous to $Enc_e$, though the convolutions are transposed. The output of $Dec$ is the point cloud, $\hat{\boldsymbol{P}}$, corresponding to the predicted coordinates of the backbone atoms, $\mathbb{V}_b \approx \boldsymbol{P}$.

**Loss Function**  The first term in the loss function is a basic reconstruction loss, where $\boldsymbol{P}$ and $\hat{\boldsymbol{P}}$ are taken to be the true and predicted coordinates of the protein backbone atoms. Namely, we evaluate their difference using Smooth-$L_1$ loss. This loss is defined, with $\delta = 2$, as

$$SmoothL_1(\boldsymbol{x}, \boldsymbol{y}) := \sum_{i=1}^{\#\boldsymbol{x}} z_i, \quad \text{where} \quad z_i = \min\left(\frac{\delta^2}{2}(x_i - y_i)^2, \delta|x_i - y_i| - \frac{1}{2}\right), \qquad (6)$$

This loss function modifies $L_2$ loss to be more robust to outliers (Girshick, 2015).

As the reconstruction loss depends on the embedding of the protein in Euclidean space, it may not best measure if intrinsic geometry is faithfully reconstructed. To address this, we consider two encoded proteins with latent codes, $[\boldsymbol{z}_{i,1}, \boldsymbol{z}_{e,1}]$ and $[\boldsymbol{z}_{i,2}, \boldsymbol{z}_{e,2}]$. Then we form a new latent variable,

$$\hat{\boldsymbol{z}}_i = (1 - \beta)\boldsymbol{z}_{i,1} + \beta\boldsymbol{z}_{i,2}, \quad \hat{\boldsymbol{z}}_e = \boldsymbol{z}_{e,1}, \qquad \beta \sim \mathbf{U}[0, 1]. \qquad (7)$$

Each of these latent variable decodes to some point cloud $\hat{\boldsymbol{P}}$. We let $Int(\hat{\boldsymbol{E}}_{t,\beta})$, $Int(\hat{\boldsymbol{E}}_{t,1})$, and $Int(\hat{\boldsymbol{E}}_{t,2})$ be the lengths of the $C_\alpha$-$C_\alpha$ pseudobonds of the generated proteins from the interpolated latent code and the two given latent codes. We then introduce a bond length penalty given by,

$$\mathcal{R}(\hat{\boldsymbol{P}}_1, \hat{\boldsymbol{P}}_2) = \mathbb{E}_\beta ||Int(\hat{\boldsymbol{E}}_{t,\beta}) - ((1 - \beta)Int(\hat{\boldsymbol{E}}_{t,1}) + \beta Int(\hat{\boldsymbol{E}}_{t,2}))||_1, \quad \beta \in \mathbf{U}[0, 1]. \qquad (8)$$

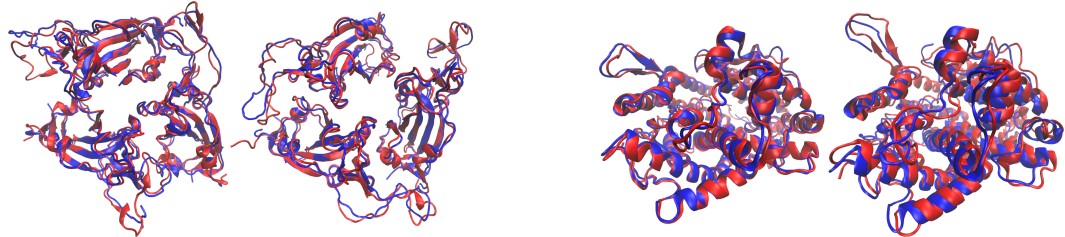

Figure 2: Reconstructions of protein frames from S protein and hACE2 test data using ProGAE. The blue and red structures correspond to the reconstructed and ground truth structures. They are superimposed for comparison. The corresponding backbones can be found in Figures 6a and 6b.

Table 1: Average atom-wise $L_2$ error, bond length error on the training and test sets, as well as RMSD between reconstructed and true structures, using ProGAE. The $L_2$ error is within the resolution of the associated PDB files, even achieving sub-Angstrom performance on human ACE2.

|  |  | **S protein** | **hACE2** |
|---|---|---|---|
| *PDB Resolution (Å)* |  | 2.68/2.80 | 2.68 |
| *Training* | Reconstruction error | $8.28 \pm 0.04$ E-1 | $4.29 \pm 0.01$ E-1 |
|  | Bond length error (Å) | $3.71 \pm 0.02$ E-1 | $1.56 \pm 0.01$ E-1 |
|  | Avg. $L_2$ error (Å) | $1.21 \pm 0.00$ | $0.78 \pm 0.01$ |
| *Test* | Reconstruction error | $1.39 \pm 0.01$ E0 | $6.27 \pm 0.03$ E-1 |
|  | Bond length error (Å) | $3.88 \pm 0.02$ E-1 | $1.63 \pm 0.02$ E-1 |
|  | Avg. $L_2$ error (Å) | $1.56 \pm 0.01$ | $0.90 \pm 0.01$ |
|  | RMSD from test set (Å) | $2.54 \pm 0.56$ | $1.24 \pm 0.23$ |

This penalty can be viewed as promoting faithful reconstruction of the pseudobond length between $C_\alpha$ atoms, as well as a smooth interpolation of these lengths along paths in $L_I$, that is independent of $L_E$. This penalty is analogous to the metric preservation regularizer introduced in (Cosmo et al., 2020) for 3D meshes. Thus, the loss function $\mathcal{L}$ for ProGAE is,

$$\mathcal{L}((\hat{\boldsymbol{P}}_1, \hat{\boldsymbol{P}}_2), (\boldsymbol{P}_1, \boldsymbol{P}_2)) := \sum_{i=1}^{2} SmoothL_1(\hat{\boldsymbol{P}}_i, \boldsymbol{P}_i) + \lambda_R \mathcal{R}(\hat{\boldsymbol{P}}_1, \hat{\boldsymbol{P}}_2). \tag{9}$$

## 3 EXPERIMENTAL SETUP

In this section, we describe the setup of our numerical experiments that confirm the usefulness of ProGAE in generating the protein conformational space. For each dataset, we train three models, each from a different random seed, and report both mean and standard deviation in our results. Details on the choice of network hyperparameters can be found in the Appendix A.3.

**Datasets** Datasets used in this work are atomistic simulation trajectories [1] (D.E. Shaw Research, 2020). The two main datasets we use are simulations of proteins in presence of FDA approved or under-investigation molecules, as we aim to test the performance of ProGAE on capturing drug-induced structural variations. These datasets are: (1) 50 independent trajectories, each simulating the SARS-CoV-2 trimeric spike protein (S protein) in the presence of a distinct drug for $2\mu$s. The simulation is limited to 3 receptor binding domains (RBDs) of the protein, as well as a short region needed for the system to maintain a trimer assembly; (2) 75 independent trajectories, each simulating the ectodomain protein of human ACE2 (hACE2) in the presence of a distinct drug for $2\mu$s.

The backbones of the S protein and the hACE2 protein contain 3,690 atoms and 2,386 atoms, respectively. The time resolution is 1,200 ps. We use the first 70% of frames from each trajectory to

---

[1]available here: http://www.deshawresearch.com/resources_sarscov2.html

Table 2: The leading canonical correlation between the learned intrinsic and extrinsic latent spaces, as well as the performance of a linear model trained on disentangled latent spaces of ProGAE for bound drug classification, are shown.

|  | S protein | hACE2 |
| --- | --- | --- |
| First canonical correlation | $0.08 \pm 0.00$ | $0.07 \pm 0.01$ |
| Drug classifier on intrinsic latent space | $2.04 \pm 0.04\%$ | $1.43 \pm 0.01\ \%$ |
| Drug classifier on extrinsic latent space | $99.65 \pm 0.26\%$ | $99.62 \pm 0.21\ \%$ |

Table 3: Results of linear regression on the extrinsic latent space for predicting physical and chemical properties of the drugs that a protein is bound to. Error is normalized for interpretability. For comparison, performance of linear regression on the PCA embeddings of the orientation of the backbone bonds is reported. This embedding is restrained to the same dimension as the latent space.

| Dataset |  | Molecular weight | Hydrogen bond donor count | Topological polar surface area |
| --- | --- | --- | --- | --- |
| S protein | PCA error ($\sigma$) | $0.78 \pm 0.00$ | $0.81 \pm 0.01$ | $0.79 \pm 0.00$ |
|  | Latent error ($\sigma$) | $\mathbf{0.55 \pm 0.04}$ | $\mathbf{0.56 \pm 0.03}$ | $\mathbf{0.61 \pm 0.00}$ |
| hACE2 | PCA error ($\sigma$) | $0.71 \pm 0.00$ | $0.65 \pm 0.00$ | $0.73 \pm 0.00$ |
|  | Latent error ($\sigma$) | $\mathbf{0.55 \pm 0.01}$ | $\mathbf{0.57 \pm 0.01}$ | $\mathbf{0.53 \pm 0.02}$ |

form the training set. The next 10% and the last 20% of frames form the validation and test sets. The train and test sets are intentionally kept temporally disjoint to better assess generalization.

For transfer learning, we also consider two trajectories of the entire S protein containing 13,455 backbone atoms. One trajectory is initiated from a closed state, while the other from a partially open state. We use the first 2.5 $\mu s$ of these 10 $\mu s$ simulations, corresponding to 2,001 frames with a resolution of 1,200 ps. Additionally, we utilize the first 10 $\mu s$ of a 100 $\mu s$ simulation of the main Protease of SARS-CoV-2, a sequence of 10,001 frames with a 1,000 ps resolution.

## 4 RESULTS

**Structure Reconstruction** Figure 2 displays the ability of ProGAE to accurately reconstruct protein conformations. The backbones are visible with atom-wise error in Figures 6a and 6b in the appendix. From the visualized atom-wise $L_2$ reconstruction error, it is clear that our network can capture and reconstruct notable conformational changes of a protein. Figures 8a and 8b in the appendix display these reconstructions with color denoting fragment instead of $L_2$ error for clarity. Consistent with the low RMSD error, visually the reconstructed structures appear consistent with ground truths, with larger RMSDs observed in the flexible loop and turn regions.

Table 1 contains performance metrics of ProGAE on training and test sets. Generalization is measured by the $L_2$ reconstruction error of the backbone atom coordinates, as well as RMSD (root mean square distance) after alignment. For hACE2, we achieve sub-Angstrom performance on the test set. In either case, the RMSD of the reconstruction is within the experimental resolution of the associated PDB files; 6VXX/6VW1 for the S protein and 6VW1 for hACE2. Additionally, the average error in the length of the pseudobonds is also sub-Angstrom. Thus, it is evident that ProGAE is able to reconstruct proteins within meaningful resolution.

**Utility of the Extrinsic Latent Space** With the reconstruction capabilities of ProGAE verified, we consider the benefit of having separate intrinsic and extrinsic latent spaces. First, we explore the statistical relationship between the learned intrinsic latent space and the extrinsic latent space. Canonical correlation analysis (CCA) is a natural approach to assess if a linear relationship exists (Hardoon et al., 2004). We include background on CCA in Appendix A.2.

Table 2 includes the leading correlation between the intrinsic and extrinsic latent spaces for each dataset. Note this correlation is very low, implying that there is a negligible linear relationship be-

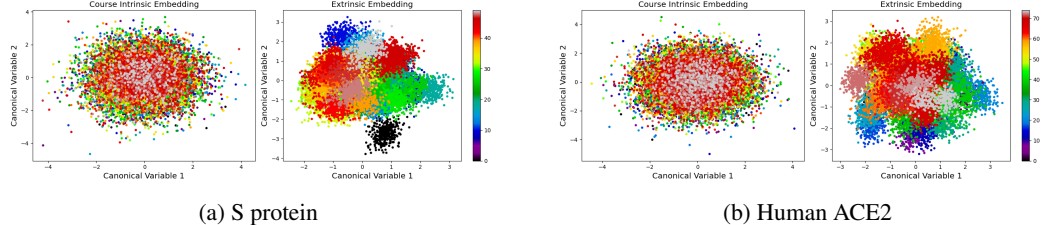

(a) S protein                    (b) Human ACE2

Figure 3: The embedding of the data in the projection of the latent space to the first two canonical vectors between the intrinsic and extrinsic latent space. Color indicates the identity of the drug that the protein is bound to within that frame. Clustering by drug identity is apparent in the extrinsic latent space, but not the intrinsic latent space. This aligns with results in Table 2.

Table 4: The percentage of backbone covalent bonds that are 10% shorter than the minimum seen in training data. The difference (Diff) between the intrinsic+extrinsic ProGAE and the extrinsic only ProGae is also reported; negative difference indicate higher bond-length error in the later model.

| Dataset | | C-CA | C-N | C-O | CA-N | CA-CA |
|---------|--|------|-----|-----|------|-------|
| S protein | Int.+Ext. (%) | 14.41 | 22.58 | 27.00 | 15.24 | 15.33 |
| | Ext. Only (%) | 19.04 | 27.34 | 27.66 | 18.58 | 20.89 |
| | Diff | -4.63 | -4.76 | -0.66 | -3.40 | -5.56 |
| hACE2 | Int.+Ext. (%) | 2.45 | 8.19 | 12.07 | 4.62 | 0.51 |
| | Ext. Only (%) | 4.99 | 9.81 | 12.34 | 5.21 | 1.59 |
| | Diff | -2.54 | -1.62 | -0.27 | -0.59 | -1.08 |

tween the intrinsic and extrinsic latent spaces. Learning a disentangled latent representation is often desired for better interpretability, which is typically measured using a generalization of mutual information such as total correlation as in (Chen et al., 2018). While structural conditions may prevent the intrinsic and extrinsic embeddings from being completely independent, Table 2 indicates absence of a linear relationship between intrinsic and extrinsic latent vectors in our learned model, confirming a notable level of disentanglement that has been explicitly encoded in our model architecture.

As stated earlier, each simulation trajectory in the dataset corresponds to the S or hACE2 protein bound to a specific drug. Then it is natural to investigate if this distinct drug information is encoded in the two disentangled latent spaces. Table 2 contains the performance of a linear classifier trained on the different latent spaces to classify the drug present in each frame. It is clear that the drug molecule can be almost perfectly classified in the extrinsic latent space, while such classification is random in the intrinsic latent space. Figures 3a and 3b display the embeddings of the test set in the latent spaces, projected to the first two canonical components. In these figures, color denotes the identity of the drug that the protein is bound to. Even in the 2D projection of the extrinsic latent space, clustering by the drug identity is apparent, which is not the case for the intrinsic embedding.

Next, we consider if this linear separation is chemically meaningful. We train a linear regression model on the extrinsic latent space to predict physico-chemical properties of a drug binding to a protein. Table 3 displays the performance of the model at predicting the properties of molecular weight, hydrogen bond donor count, and topological polar surface area. For comparison to our latent embedding, we train a linear regression model on the first $n_e$ principal component scores of the PCA of the extrinsic signal on each element of the test dataset. The latent regression outperforms that of PCA, indicating that the the extrinsic latent embedding captures more physico-chemical information about the bound drug. We believe this linear regression is appropriate as it prevents overfitting.

**Utility of the Intrinsic Latent Space** Having confirmed the utility of a separate extrinsic latent space, we weigh the benefits of including the intrinsic latent space in the model. This is nontrivial, as the mean pseudo-bond length is 3.86Å with a deviation of 0.06. We find the inclusion of the intrinsic latent space improves the geometric validity of the reconstructed protein as seen in the

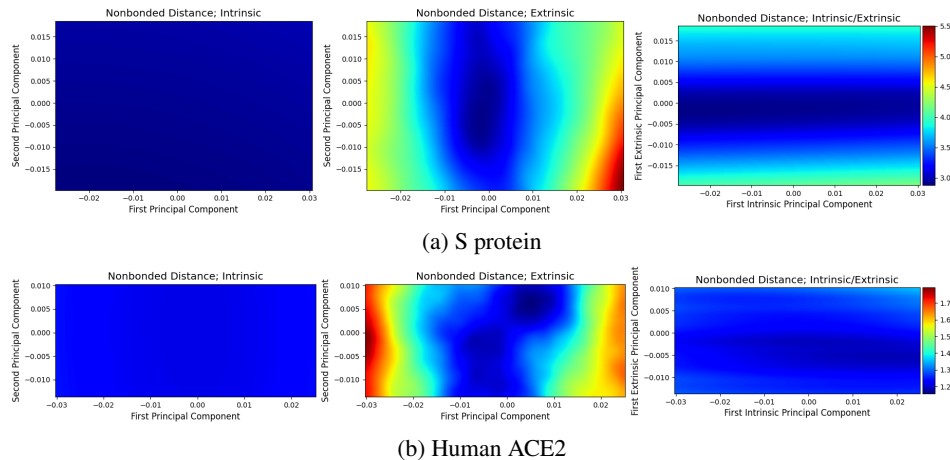

(a) S protein

(b) Human ACE2

Figure 4: Error of the non-bonded distance between the backbone atoms for generated structures compared to the protein at its initiation. Each point corresponds to a structure generated by sampling the latent space via the plane of the two specified principal components of the latent spaces. The intrinsic latent code hardly affects the non-bonded distance error, unlike the extrinsic latent code.

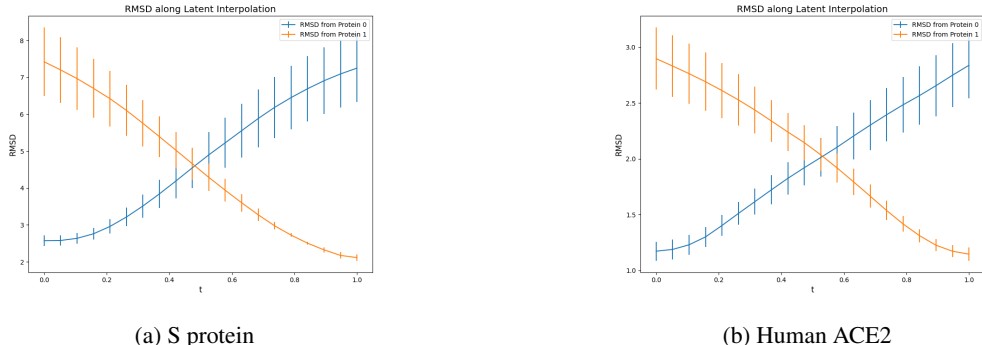

(a) S protein

(b) Human ACE2

Figure 5: RMSD of proteins generated along the latent interpolation between two proteins from different trajectories. The RMSDs are computed with respect to the endpoint proteins, with mean and standard error shown. We see a smooth interpolation between the RMSD errors as desired. Examples of the structures along the interpolation path can be found in the appendix in Figure 7.

following ablation study. We trained a model that only encodes the extrinsic signal to reconstruct the protein. While it was comparable in performance regarding $L_2$ error, we found this extrinsic-only model resulted in a higher percentage of erroneous bonds. This is shown in Table 4. Here we define a erroneous bond, if the bond length deviates by more than 10% from the minimum of the ground truth distribution, as such deviations will result in steric clashes. We also trained a model on only the intrinsic signal, but did not analyze the model further due to poor performance ($> 3.0$Å). These results are found within the appendix in Table 6.

**Geometry of the Generated Structural Ensemble**    As a generative model, it is important to consider if paths in the latent space meet our expectations. One way by which we measure this is how the pairwise distance matrix between non-bonded atoms in a generated protein changes as the latent variables change. Specifically,we are interested in the distance between non-bonded atoms.

In Figures 4a and 4b, we plot the norm of the difference between these distance matrices for the generated protein and that of the protein at its initiation. We sample the latent space along two principal components, while setting others to 0. So in these figures 4a and 4b, we see three different cross sections of this space. It is immediately apparent that a change in the extrinsic latent code dominates changes in the non-bonded distance. This is a desired behavior of a disentangled geometric

Table 5: Average atom-wise $L_2$ error in reconstruction (Å) on the test dataset after transferring a trained model to other protein structures. These simulations correspond to two different states of the entire S protein and of the main protease of SARS-CoV-2. We retrain only the dense layer mapping to the decoder. For comparison, we train the same layer for a randomly initialized model (Baseline).

| Source dataset | **Closed S** | **Partially Open S** | **Protease** |
|---|---|---|---|
| Baseline (Å) | $1.79 \pm 0.04$ | $2.26 \pm 0.03$ | $1.36 \pm 0.03$ |
| S protein (Å) | $1.55 \pm 0.01$ | $2.01 \pm 0.07$ | $\mathbf{1.16 \pm 0.00}$ |
| hACE2 (Å) | $\mathbf{1.52 \pm 0.03}$ | $\mathbf{1.98 \pm 0.00}$ | $1.17 \pm 0.01$ |

autoencoder, where large-scale changes are captured in the extrinsic latent space, while small-scale changes in local distance are controlled by the intrinsic latent space.

We also evaluate the performance of linear interpolations in the learned latent space. Given two protein conformations from different trajectories (i.e. in the context of two different drugs), we generate a path between them by generating the linear interpolation of their latent codes. This provides a path of structural variation that does not exist in the training data. The results of this interpolation in terms of RMSD is shown in Figure 5. As expected, we see a smooth exchange in the RMSD error of the generated protein from the first protein and from the second protein.

**Transfer Learning– Extension to Different Proteins**    To check the generalization of ProGAE, we investigate transfer learning to simulations of different proteins. We begin with models trained on the S protein comprised of the 3 RBDs and on hACE2. These results are summarized in Table 5. We transfer learned ProGAE models to trajectories of the closed and partially open state of the entire S protein, as well as SARS-COV-2 main Protease, which provides insight into the generalization capability of the convolutional filters that we have learned. As a result, six scenarios of the transfer learning, in addition to three random baselines, are reported in Table 5. When transferring the model trained on the 3 RBDs of the S protein to the S protein in the closed state, we are transferring the model learned on a partial structure to the entire protein *that is much larger in size*. Model transfer to the S protein in the partially open state deals with a scenario where *the conformational state of the protein is notably different (closed vs partially open)*. Transferring the model trained on hACE2 to the S protein datasets studies the knowledge transfer to an *entirely different protein, but one which hACE2 is known to interact with*. Finally, transferring both S protein and hACE2 models to the main Protease simulation allows us to study the transfer of the models to a *completely different protein without notable interaction with the source protein*. Given performing long time-scale simulations of large protein systems at high resolution is a computationally expensive process, our method appears beneficial, as ProGAE transfers well to non-related proteins of larger size.

The only incompatible layer is the dense layer mapping from the latent spaces. To investigate transfer learning, we train just this dense layer for 10 epochs. As a baseline, we train the same layer of a randomly initialized model. In all cases, the transferred model performs better than the baseline. Thus the learned filters generalize to trajectories of completely different protein systems.

## 5    CONCLUSION

In this work we introduce a novel geometric autoencoder named ProGAE for learning meaningful disentangled representations of the protein conformational space. Our model accurately reconstructs protein structure. The autoencoder separately encodes intrinsic and extrinsic geometries to ensure better latent space interpretability. The extrinsic latent space can classify the protein structures with respect to the bound drug molecules, as well as can predict the drug properties. The intrinsic latent space assists in improving the validity of the bond geometry in the reconstructions. We also show that the geometric convolutional filters learned in training can be successfully transferred to trajectories of different protein systems, irrespective of system size, conformational state, or presence of protein-protein interaction. These results on learning, predicting, and generating protein conformations suggest that the proposed framework can serve as the first step towards bridging geometric deep learning with molecular simulations.

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
