# OpenReview forum: "ProGAE: A Geometric Autoencoder-based Generative Model for  Disentangling Protein Conformational Space"
_ICLR.cc/2021/Conference — Reject_

### Official Review · AnonReviewer3 · 2020-10-26
**This paper presents a method of capturing latent embedding from 3D protein structure training data and reconstructing protein coordinates for test data. A point is to separate embeddings for intrinsic (alpha carbon connections) and extrinsic (bond orientation) properties.**

**Rating:** 5
**Confidence:** 2

**Review:**

The application is a very interesting topic in general, and in this sense the work would be important, while the organization of this paper is not good enough, particularly the method section, which should be written in a more organized manner. Or currently the method section is written rather too plainly and wordy, while I think a more easy-to-read  organization would be needed with illustrations. Otherwise it is hard to see what are the new points in details by this paper. I have the following as other comments:

1. In fact the method is somewhat different from (what expected from) that described in Introduction. This is because "dynamics" means time-series, by which I thought the latent embeddings are also time-series (like exactly simulation), while the reality is slightly different.

2. Again the method section particularly encoding should be more clearly described.

3. Datasets are limited to only one, two types of proteins. I'm not familiar with the field, while as a methodology would it be good enough only checking these small number of proteins? I think it would be hard to say something general only from such a small number of proteins.

4. It is not clear how each of the intrinsic and extrinsic properties contributed to the prediction. The performance results obtained by each of them should be shown.

5. Also evaluation would not be so strict, in the sense of no comparison with other methods, except the baseline in transfer learning.

Other minor points:
1. I think the authors explain what "intrinsic and extrinsic properties" mean somewhere earlier in brief.

2. This paper has a lot of typos: "After that, We discuss", "that do are referred", etc.

---

> ### Author Response · Authors · 2020-11-25
> **Thank you for your feedback**
>
> Q:”This is because "dynamics" means time-series, by which I thought the latent embeddings are also time-series (like exactly simulation), while the reality is slightly different.”
>
> Ans:
> As mentioned to other reviewers, we have avoided using the phrase “generating dynamics” in the revision. While we train our ProGAE model on protein structures generated using long time-scale molecular dynamics simulations, the goal is to learn the protein conformational space and generate valid samples from that space. Results show that ProGAE provides a smooth latent embedding of the conformational space (Figure 5). This is  beneficial to enhance sampling of larger protein systems that are computationally expensive to simulate, as shown in the transfer learning experiments (Table 5). Additionally, we assess the generalization of the model via a test set that is temporally disjoint from the training set.
>
> ---
>
> Q:”Again the method section particularly encoding should be more clearly described.”
>
> Ans:
> We have revised the paper to make the encoding of the network more clear. We have included equations 4 and 5 to more explicitly formulate the extrinsic encoder outside of the paragraphs for clarity.
>
> ---
>
> Q:”Datasets are limited to only one, two types of proteins. I'm not familiar with the field, while as a methodology would it be good enough only checking these small number of proteins? ”
>
> Ans:
> Regarding the number of datasets considered in this paper, our goal is to learn to model the structural variations present in  the simulated conformational landscape of a biologically relevant protein. For that purpose, we needed the model to be trained on long time-scale (microsecond scale) simulations of a protein,  which is computationally expensive to generate and not available to scale. We train our model on  50 and 75 different trajectories, each trajectory capturing binding to a different drug molecule, for two different proteins that are biologically relevant in terms of system size and reported function. The end goal is to test  our method on how well it captures distinct protein structural changes associated with drug binding, which is of critical importance for drug discovery.  We conducted the transfer learning experiment to confirm that the convolutional filters learned in the single protein case can be transferred to different-size and  unrelated proteins or to protein conformations corresponding to different states (open and close).
>
> ---
>
> Q:”It is not clear how each of the intrinsic and extrinsic properties contributed to the prediction. The performance results obtained by each of them should be shown.”
>
> Ans:
> We have included an ablation study showing how the intrinsic and extrinsic components contribute to the model. The extrinsic component is necessary for reconstruction, while the intrinsic component improves the validity of the reconstructed proteins. This can be seen in Table 4 where we compare the percentage of bond lengths that are unrealistically small resulting from a model trained on both intrinsic and extrinsic data to a model trained solely on extrinsic data. The model trained solely on extrinsic data results in more of these erroneous bonds. These small bond lengths are responsible for steric clashes and thus invalid bond geometries.
>
> ---
>
> Q: “Also evaluation would not be so strict, in the sense of no comparison with other methods, except the baseline in transfer learning.”
>
> Ans:
> As mentioned to another reviewer, we now discuss in the Related Work section why we lack comparisons to other methods. The 3 reasons include lack of available code or pretrained models, a method does not generate 3D backbone atom coordinates, or the related model is not scalable to our case of large-scale protein.

---

### Official Review · AnonReviewer1 · 2020-10-28
**Official blind review #1**

**Rating:** 7
**Confidence:** 4

**Review:**

**Summary**
The paper introduces a geometric variational autoencoder for capturing protein structural ensembles, disentangling intrinsic and extrinsic geometry into separate latent spaces. The model is shown to accurately reconstruct protein structure, and the difference between the intrinsic and extrinsic latent spaces are explored. Finally, the model is tested in a transfer-learning setting, where it displays encouraging results.

**Strengths**
Learning representations of protein structure is an important problem, which has only received limited attention in the literature so far. The paper presents a new approach for tackling this problem, and reports good results. The paper is thus interesting both from a methodological and an application perspective.

**Weaknesses**
I only identified minor weaknesses, which I address in detail below.

**Recommendation**
I recommend the paper be accepted. The presented method is novel (to my knowledge), and it is clearly presented.

**Comments to the authors**
Although you give a clear definition of extrinsic vs intrinsic geometry, it would be enlighening if a slightly more elaborate motivation was given for why this is a meaningful way to separate the geometric properties of a protein. Which structural properies to we expect to capture in each of the two distinct latent space? Perhaps you could relate this to the discussion in protein modelling of whether to represent protein structure using internal coordinates (angles, dihedrals, bond-angles) vs the coordinates of all atom positions. In the result section, you demonstrate a case where the extrinsic latent space separates structural changes in response to drug changes, but are there any cases where the intrinsic representation is a useful descriptor?

Page 3, "we note that the protein backbone trace can be viewed...The protein backbone itself can be viewed." These two sentences are central to the paper, but not entirely clear.  The first refers to the "backbone trace" and the other to the "backbone itself", but the distinction between these is not formally defined (at least as far as I could see). Perhaps it is merely a question of defining what the "trace" of a protein is.

Several places in the manuscript you refer to "generating protein dynamics", and "reconstructing protein trajectories". As far as I could see from the paper, the model only describes how single structures can be encoded and decoded, making it a probabilistic model over structure. My interpretation is therefore that it is a model that can generate structural ensembles - i.e. reconstructing thermodynamics, but not dynamics or trajectories, which would seem to me to require the definition of some process in the latent space. Please either rephrase these places in the text, or make it more clear how trajectories are generated.

In your final transfer learning experiment, you conclude that "the learned fiters generalize to trajectories of completely different protein systems". Have you looked into how the representations are effected. More specifically, do you see a difference in how well the intrinsic vs the extrinsic features generalize?

**Minor comments**
Page 1, "and improving latent" -> "and improve latent"

Page 1, "intrinsic and extrinsic geometries". The introduction would be slightly easier to follow if the difference between intrinsic and extrinsic geometries were explained informally already very early in the paper - so that we wouldn't have to wait until the bottom of page 2 before these terms are defined.

Figure 1. The left most figures (the protein structures) do not very clearly illustrate the difference between intrinsic and extrinsic geometry. Could this difference perhaps be made clearer by zooming in to part of the structure?

Figure 1, caption: "encodeing" -> "encoding"

Page 3, "where as backbond" -> where as backbone"

---

> ### Author Response · Authors · 2020-11-25
> **Thank you for your feedback**
>
> Q:”but are there any cases where the intrinsic representation is a useful descriptor?”
>
> Ans:
> As mentioned in the responses to other reviewers, we have included an ablation study to check the effect of encoding intrinsic only, extrinsic only, and intrinsic+extrinsic signals in the model on the MSE loss, and more importantly, on the bond lengths in the reconstructed proteins (reported in Table 4). We find that a model trained on both intrinsic and extrinsic signals results in a smaller percentage of short covalent bonds than a model trained  only on the extrinsic signal. Thus the intrinsic latent code is useful for increasing the validity of reconstruction and avoiding steric clashes.
>
> ---
>
> Q:”Perhaps it is merely a question of defining what the "trace" of a protein is.”
>
> Ans:
> We have more explicitly defined the protein trace and protein backbone in the paper. The protein trace refers to just the set of CA atoms and their (pseudo) connectivity. It is a coarser description than the protein backbone, where the backbone contains all atoms on the amide planes (except hydrogen) which are joined via CA atoms.
>
> ---
>
> Q:”...not dynamics or trajectories, which would seem to me to require the definition of some process in the latent space. Please either rephrase these places in the text, or make it more clear how trajectories are generated.”
>
> Ans:
> We appreciate the suggestion on avoiding the phrase “generating dynamics”. As such we have rephrased our usage in the paper. We train a model on molecular simulation trajectories to learn and sample from the protein conformational space. Our training data and test data are temporally disjoint.
>
> ---
>
> We appreciate the other comments made by the reviewer and addressed them in the revised manuscript.

---

### Official Review · AnonReviewer4 · 2020-10-28
**An interesting application, but the paper looks underdeveloped**

**Rating:** 5
**Confidence:** 4

**Review:**

The authors address a problem of learning protein dynamics directly from the protein structures. They propose an autoencoder type approach with disentanglement intrinsic and extrinsic features. However, there are several concerns with this approach:

1. Novelty - it is a certainly novel application with an adaptation to proteins of the paper from Tatro et al., but the papers look very similar. Could the authors please elaborate more on the methodological differences between these two approaches? Is there any novelty that I missed except that the idea was applied to another domain?
2. “Disentanglement” is not learned by the model, but actually hard coded by expert knowledge. It is not surprising at all that two spaces are disentangled at all - the authors just fed two different types of manually crafted representations from the very beginning. I am not sure that it can count as methodologically new disentanglement from ML point of view.
3. The idea of intrinsic and extrinsic components is nice, however from the experiments the contribution of the intrinsic component to the reconstruction is not very clear.
4. The results from Table 3 are indeed interesting, but they don’t mean that “the extrinsic latent embedding captures more physicochemical information about the bound drug”, they just mean that the relationship between raw extrinsic features to the drug properties is rather non-linear. As a fair comparison authors could rather use a non-linear model (e.g. neural network) for the same task instead of PCA.
5. The authors use reconstruction metric to evaluate their model performance, which is reasonable for model comparisons, but I am not sure that this is an indicator that the proposed model and the approach could be used in practice. How will the authors sample protein conformations from latent trajectories? Maybe the authors could elaborate on it more and provide examples/experiments? For example, Figure 4 shows how distances change when they sample along principle component, but it doesn't mean that reconstructed conformations are meaningful or realistic. Otherwise, the only thing I see is: it is possible to train an AE type of model (even with heavily predefined features as input). So summing 4 and 5 together it would be interesting to see that the model is indeed useful and goes beyond non-linear regression.
6. The authors cite several related methods (“Specifically, there has been interest in modeling the underlying conformational space of protein dynamics via deterministic… and probabilistic…”), however there is no comparison to these methods in the paper. Would be nice if the authors could be add it or at least explain why they don't.

---

> ### Author Response · Authors · 2020-11-25
> **Thank you for your feedback**
>
> Q:”Novelty - it is a certainly novel application with an adaptation to proteins of the paper from Tatro et al., but the papers look very similar. Could the authors please elaborate more on the methodological differences between these two approaches? Is there any novelty that I missed except that the idea was applied to another domain?”
>
> Ans:
> The paper by Tatro et al. focuses on regular surfaces. While the present approach is motivated by that work, it is not straightforward to define the intrinsic and extrinsic signals on a protein which we model as a graph. Further, we operate on the level of two different geometric scales, coarse and fine. The intrinsic signal is on a coarse scale, the protein trace, and the extrinsic signal is on a finer scale, the protein backbone. The selection of these objects results from domain expertise that informs the model design. The use of multiscale geometry distinguishes this work from Tatro et al.
>
> ---
>
> Q:” I am not sure that it can count as methodologically new disentanglement from ML point of view.”
>
> Ans:
> Regarding disentanglement, it is not necessarily guaranteed that the intrinsic and extrinsic latent space will be disentangled just because they are the results of separate encodings. This is because the input signals could be correlated, which is not found in the proposed framework as seen in Table 2. This is one reason why we introduce the pseudo-bond length penalty; to promote invariance in the intrinsic latent variable compared to that of the extrinsic one. Designing the disentanglement by including prior domain knowledge provides better performance and interpretability.
> ---
>
> Q:”The idea of intrinsic and extrinsic components is nice, however from the experiments the contribution of the intrinsic component to the reconstruction is not very clear.”
>
> Ans:
> As noted in the response to the other reviewers, we include an analysis of the bond lengths after reconstruction in Table 4. The model trained on both intrinsic and extrinsic signal results in less unrealistic shorter bonds, when compared to the model trained solely on extrinsic signal. Such short bonds are responsible for steric clashes, resulting in geometrically erroneous structures.
>
> ---
>
> Q:”The results from Table 3 are indeed interesting, but they don’t mean that “the extrinsic latent embedding captures more physicochemical information about the bound drug”, they just mean that the relationship between raw extrinsic features to the drug properties is rather non-linear. As a fair comparison authors could rather use a non-linear model (e.g. neural network) for the same task instead of PCA.”
>
> Ans:
> Regarding Table 3, we wanted to compare the learned latent space to some other embedding of the protein dataset. The PCA of the proteins seemed like a natural comparison. We conduct linear regression on these embeddings, as it provides a simple analysis of the embeddings and avoids overfitting.
>
> ---
>
> Q:”How will the authors sample protein conformations from latent trajectories?”
>
> Ans:
> In the revision, we more closely consider sampling from latent trajectories.Figure 7 shows the latent path between two proteins as a simple linear interpolation. Figure 5 displays the RMSD of proteins generated along this interpolation to the endpoint proteins.
>
> ---
>
> Q:”However there is no comparison to these methods in the paper. Would be nice if the authors could be add it or at least explain why they don't.”
>
> Ans:
> In the Related Work section, we discuss why we do not have comparisons to related work. There are 3 main reasons why we lack comparisons. Commonly, there was no code or pretrained models available. Additionally, a body of related work was concerned with the generation of the protein contact map and not the direct reconstruction of the backbone atom coordinates. Finally, one method had code available but used networks that did not scale to our case, as we work with proteins that have over 900 residues.

---

### Official Review · AnonReviewer2 · 2020-10-29
**Method for analysis of MD trajectories; Needs further evaluation**

**Rating:** 4
**Confidence:** 4

**Review:**

This paper presents a method for reconstructing trajectories of protein structure from MD simulation. The main technical contribution is a geometric autoencoder architecture with separate latent spaces for representing intrinsic and extrinsic geometry.

Overall, while the latent space design is interesting, the evaluation of the method is not totally convincing, and there seems to be a lot left on the table in terms of extracting dynamics information from the data. The primary goal of the method appears to be analysis of MD simulations, however the learned representations are compared to quantities that are very easy to directly measure, so I’m not convinced that the method would be useful for analysis in an unsupervised setting. The validity of the reconstructed structures should be more thoroughly characterized.

Questions/comments:
- The model appears to be trained on static snapshots from the trajectory — are dynamics/temporal information included in any way?
- To test the usefulness of the different intrinsic/extrinsic representations (the main claim of this paper), the authors should perform an ablation study, removing the either the extrinsic or intrinsic components.
- The intrinsic geometry is represented by Ca-Ca bond distances. These backbone distances typically stay very close to 3.86 A in crystal structures due the chemical constraints from the peptide backbone. Do they vary in the simulation - does it really need to be learned?
- While L2 loss provides a global measure of the reconstruction error, it would be interesting to report additional metrics on the validity of the reconstructed structures, e.g. steric clashes, invalid bond geometries.
- It is not clear to me from the visualization in Fig 3 that the drug molecule is “perfectly classified” in the latent space.
- Can the authors clarify how transfer learning works when transferring the trained model to another molecule with a different number of atoms? What is the purpose of transfer learning? It would be useful to include a baseline of a proGAE model trained from scratch on the S/Protease trajectories in Tab 4.
- For the transfer learning task, the authors should show visualizations of the reconstruction vs. ground truth as in Fig 2. Again, it is hard to interpret the aggregate MSE error, and I would be interested to see how the reconstruction compares between the different source models.

---

> ### Author Response · Authors · 2020-11-25
> **Thank you for your feedback**
>
> Q: “The model appears to be trained on static snapshots from the trajectory — are dynamics/temporal information included in any way?”
>
> Ans:
> Since the goal of this work is to capture and generate structural variations present in protein simulations, to avoid confusion we revise the text all throughout.
> Nevertheless, temporal information is integrated into the model through training. Namely, the model is trained on the first 70% of frames in each trajectory, whereas the validation and test sets comprise the last 30% of frames. So while the model is trained on static frames, the training set and test set are temporally disjoint.
> ---
>
> Q: “To test the usefulness of the different intrinsic/extrinsic representations (the main claim of this paper), the authors should perform an ablation study, removing the either the extrinsic or intrinsic components.”
>
> Ans:
> We have included additional analysis of the validity of reconstructed structures for different models in an ablation study, namely a model trained on extrinsic information alone, a model trained on the intrinsic information alone,  and the model that uses both intrinsic and extrinsic information. The model trained solely on intrinsic data is not able to learn to minimize L_2 reconstruction loss. While the model trained solely on extrinsic data can learn to minimize L_2 reconstruction loss, it produces reconstructions with a higher percentage of unrealistically small bond lengths compared to the model with intrinsic and extrinsic components.
> In Table 4 we analyzed the percentage of bonds with  small enough length to be considered erroneous with respect to the test data in reconstructed structures. This metric is useful to quantify geometric error (i.e. steric clashes) present in the reconstructed structures. Table 4 shows that  encoding intrinsic information helps alleviate bond length error.
>
> ---
>
> Q: “The intrinsic geometry is represented by Ca-Ca bond distances. These backbone distances typically stay very close to 3.86 A in crystal structures due the chemical constraints from the peptide backbone. Do they vary in the simulation - does it really need to be learned?”
>
> Ans:
> The CA-CA bond distances are tightly distributed around 3.86 A, but there is still some variation, roughly 0.057 A. These are seen to vary in simulations as expected , as they are not fixed like the bond lengths. From our ablation study concerning bond lengths, we find that the integration of CA-CA pseudobond length reduces the number of steric clashes in the reconstructions. Thus, the intrinsic information helps to improve the validity of reconstructions.
>
> ---
>
> Q: “While L2 loss provides a global measure of the reconstruction error, it would be interesting to report additional metrics on the validity of the reconstructed structures, e.g. steric clashes, invalid bond geometries.”
>
> Ans:
> As suggested, we have provided information related to the validity of the bond geometries. Table 4 provides information related to steric clashes present in the reconstructed proteins. Additionally, we have included RMSD of the aligned reconstructed protein and ground truth protein for the test set found in Table 1. This provides a more standard measure than L_2.
>
> ---
>
> Q: “It is not clear to me from the visualization in Fig 3 that the drug molecule is “perfectly classified” in the latent space.”
>
> Ans:
> Regarding Figure 3, there is notable clustering of drug identity in the 2D projection of the extrinsic latent space. This figure complements Table 2, where the linear classifier achieves near 100% accuracy. We make the connection between this table and figure more clear in the revision.
>
> ---
>
> Q:”Can the authors clarify how transfer learning works when transferring the trained model to another molecule with a different number of atoms? What is the purpose of transfer learning? It would be useful to include a baseline of a proGAE model trained from scratch on the S/Protease trajectories in Tab 4.”
>
> Ans:
> Regarding transfer learning when the proteins have a different number of atoms; this requires a change in the graph connectivity. This change in the graph does not prevent us from using the same graph convolutional filters since graph attention filters are functions on a single vertex or edge. The dense layers mapping to and from the latent space need to be reshaped though. This is precisely the layer trained from scratch during transfer learning, while convolutional layers are fixed. The purpose of the transfer learning is to demonstrate that the convolutional filters learned are meaningful enough to at least be used as a good initialization for training. This is why we compare to fixed random convolutional filters.

---

### Decision · Program_Chairs · 2021-01-07
**Final Decision**

**Decision:**

Reject

**Comment:**

This paper addresses an interesting learning problem of a generative neural network on a simulated ensemble of protein structures obtained using molecular simulation to characterize the distinct structural fluctuations of a protein bound to various drug molecules. The main technical contribution is a geometric autoencoder architecture with separate latent spaces for representing intrinsic and extrinsic geometry. However, the reviewers think the benefit for modeling intrinsic and extrinsic geometry is not clearly explained and the experiments are not convincing at the moment. The paper can be potentially improved by addressing these two main issues.